# Interpretable Assessment of ST-Segment Deviation in ECG Time Series

**DOI:** 10.3390/s22134919

**Published:** 2022-06-29

**Authors:** Israel Campero Jurado, Andrejs Fedjajevs, Joaquin Vanschoren, Aarnout Brombacher

**Affiliations:** 1Department of Mathematics and Computer Science, Eindhoven University of Technology, 5612 AZ Eindhoven, The Netherlands; j.vanschoren@tue.nl; 2IMEC the Netherlands, Holst Centre, 5656 AE Eindhoven, The Netherlands; 3Philips, High Tech Campus 34, 5656 AE Eindhoven, The Netherlands; andrejs.fedjajevs@philips.com; 4Department of Industrial Design, Eindhoven University of Technology, 5612 AZ Eindhoven, The Netherlands; a.c.brombacher@tue.nl

**Keywords:** automated machine learning, hyperparameter optimization, electrocardiogram, time series, ST-segment, myocardial infarction

## Abstract

Nowadays, even with all the tremendous advances in medicine and health protocols, cardiovascular diseases (CVD) continue to be one of the major causes of death. In the present work, we focus on a specific abnormality: ST-segment deviation, which occurs regularly in high-performance athletes and elderly people, serving as a myocardial infarction (MI) indicator. It is usually diagnosed manually by experts, through visual interpretation of the printed electrocardiography (ECG) signal. We propose a methodology to detect ST-segment deviation and quantify its scale up to 1 mV by extracting statistical, point-to-point beat characteristics and signal quality indexes (SQIs) from single-lead ECG. We do so by applying automated machine learning methods to find the best hyperparameter configuration for classification and regression models. For validation of our method, we use the ST-T database from Physionet; the results show that our method obtains 98.30% accuracy in the case of a multiclass problem and 99.87% accuracy in the case of binarization.

## 1. Introduction

Cardiovascular diseases (CVD) are one of the most harmful health-related problems affecting society today, and unfortunately, these disorders are becoming more widespread, with the highest number (17 million) of deaths between 2007 and 2017 being associated with CVD. This means that CVD increased by 21% in 10 years, and the most prevalent condition was stroke and ischemia, to which 8.93 million deaths are attributed [1]. It is estimated that 17.9 million people will suffer some CVD by 2060 in Europe alone [2]. The healthcare industry is in need of tools to facilitate the diagnosis of CVD automatically and with high accuracy [3]. CVD can be categorized in many different forms, of which arrhythmias are the most frequent ones [2]. Prevalently, the correct diagnosis and abnormality identification are made based on features extracted from the electrocardiography (ECG) signal, such as mean heart rate (HR), heart rate variability (HRV), R-R intervals, root mean square of successive differences between normal heartbeats, etc., or specific morphology characteristics. Hence, ECG is frequently used to monitor the electrical activity and muscle contractions of the heart. ECG analysis for CVD detection is indispensable, as morphological patterns are always considered to allow experts to discriminate abnormalities. ECG analysis provides the detailed cardiovascular health status of the user, but one of the crucial challenges lies in the automation of the processing.

The continuous ECG signal can be split into repetitive segments, or beats, which themselves are characterized by a standard set of fiducial points (Figure 1). Subject to the specific positioning of ECG electrodes on the body to diagnose various CVD, these points may become visible. The ST-segment refers to the slice connecting S and T peaks, and the ST-segment deviation is defined as the relative amplitude positioning to the isoelectric line or baseline, which is described as the amplitude in 1 millivolt (mV) between the P-wave offset and the R-onset. ST-segment changes are indicators of possible arrhythmia or ischemia, and are split into ST elevation and ST depression.

The ST-segment deviation is characterized by the isoelectric line having a different value in microvolts μV after the QRS complex; see Figure 2. The classic way for the cardiologist to calculate the ST-segment change is to view the ECG data on a specialized grid; each 1 × 1 mm square on this grid corresponds to 0.04 s. Normally, the change in ST-segment is measured as 60/80 milliseconds (ms) from the J-point. ST-segment elevation is considered a significant symptom of myocardial infarction (MI) and is caused by inflammation in the vascular wall. MI can occur suddenly, without preliminary symptoms, so early identification is crucial [5]. An additional factor in the diagnosis of MI is the level of elevation, which is correlated to the degree of danger for an individual. Significantly higher infarction and hospital mortality frequently occurs when elevation exceeds 2 mV [6]. On the other hand, ST-segment depression occurs quite regularly in athletes and elderly people [7]. It is an indicator of myocardial ischemia and descending coronary artery, which also affects the morphology of the QRS complex. [8]. That is why its timely identification is of the utmost importance. The visibility of the ST-segment in a particular ECG recording is dependent on the positioning of the electrodes [9] and, therefore, its diagnosis may be overlooked, even by experts.

Consequently, it is essential to provide tools that can accurately detect ST-segment deviation and present the results to experts in the field in an interpretable and transparent manner. Previous research has used deep learning models to more accurately approximate CVD, obtaining promising results. Such is the case of [10], where they used a CNN to classify atrial fibrillation, ventricular fibrillation, and ST-segment deviation. The disadvantage of this work is that considerable computational capacity is required if ECG segments ranging from hours to days are used, as the signals are first converted to a representation that the CNNs can understand, and then the CNN itself requires more computational resources. Other examples include [11,12,13]. Rajpurkar, P. et al. decomposed the signal into its frequency components to create images and then fed them to CNN. Other works focus on feature selection and feature extraction. For example in [14], Tang X. et al. classified the QRS complex, Q wave, and T wave in an ECG signal through an algorithm that was embedded in an FPGA, where they obtained a classification accuracy of 96%. All these works can contribute to ST-segment deviation detection. However, it is necessary to implement methods that eliminate bias in choosing the combination of hyperparameters, models, and features in CVD identification, the aim being to achieve objective solutions that can evolve over time. The difference with our work is that we do feature selection and data modeling automatically through automated machine learning (AutoML).

The contribution of this work is a method for the classification of ST-segment deviation, and the detection of such abnormality in millivolts. The output of the algorithm is ECG beats classification into either normal, elevated, or depressed classes. Additionally, we conducted a regression task to give the value of the deviation in mV, since another of our contributions is to provide a solution that is interpretable and does not require a great deal of computing power to reproduce. The algorithm is able to estimate ST-segment from a single lead ECG.

This paper is organized as follows: Section 2 presents our suggested methodology and experimental setup. Section 3 shows the results and advantages of our approach. Finally, Section 4 includes a discussion of these results. We conclude by addressing the relevance of this work is compared to other work and includes our future research.

## 2. Materials & Methods

### 2.1. Overview

The general idea of our approach is visualized in Figure 3. Given an ECG time series, we separate it by each beat, and we extract detailed characteristics to form a tabular dataset. These features are used to train and test the regression and classification task with ML models through AutoML. We explain the algorithms with more detail in the following sections.

### 2.2. Database

Electrode positioning can affect ECG morphology, there are 12 ECG lead configurations known as $V_{1},\ldots ,V_{6}$, and leads I, II, III, aVL, aVR, and aVF. Lead $V_{5}$ is the most widely preferred in practice to observe ST-segment changes. We used the European ST-T Database from Physionet [15] to create and validate our models; the data were acquired from leads $V_{1},\ldots ,V_{5}$. This database is a collection of 90 ambulatory records from 79 people: 70 men, 8 women and 1 is unknown (this may bias the results, and we will use more gender-balanced datasets in future work). All subjects were suffering from a specific cardiac disease. Each record has a duration of two hours with a sampling frequency of 250 Hz. In addition, annotations of the deviation value in μV of the ST segments were made by two independent cardiologists. Cases in which the labels were different by them were corrected afterwards by consensus between the annotators.

### 2.3. Feature Selection and Feature Extraction

The ECG time series data were retrieved from the database, and we preprocessed them as follows. We cleaned the data using a high-pass Butterworth filter with a cutoff frequency of 0.5 Hz, in order to remove baseline drift and DC offset. After that, the R-peaks were located in the filtered signals using the tool described in [4]. Having each heartbeat independent, we normalized them concerning the maximum and minimum points, as seen in Equation (Equation 1):(1)beat_scaled=(beat−min(beat))/(max(beat)−min(beat))

Separate beats allow features to be extracted independently. Experts establish the importance in having analysis per beat and performing validation based on the number of consecutive beats with a certain abnormality [16]. For this point, we performed different combinations of simple statistical, or inherent features of the signal were extracted, as shown in Table 1. In the results section, we will discuss the comparison between the combination of these characteristics.

The features we are going to mention are a combination of variables used in previous work for ECG classification in different domains [17], as well as other frequently used statistical features, in order to provide an easily reproducible and robust solution.

In subset *A*, we included time–distance features calculated by subtracting timestamps of detected points of interest and the R-peak. Points of interest included the P−peaks, Q−peaks, S−peaks, T−peaks, P−onset point, P−offset point, and T−offset point.In subset *B*, we included statistical measures of the whole ECG beat, as its median, standard deviation (STD), kurtosis (kSQI, defined in Equation (Equation 3)) and skewness (sSQI, Equation (Equation 2)), known as the third (v3) and fourth (v4) standardized moments.
(2)$$sSQI=\frac{E\left\{{\left(x-\mu_{x}\right)}^{3}\right\}}{\sigma^{3}}=v^{3}$$
(3)$$kSQI=\frac{E\left\{{\left(x-\mu_{x}\right)}^{4}\right\}}{\sigma^{4}}=v^{4}$$$\mu_{x}$ and σ are the mean and standard deviation of the signal, respectively. As well as the power spectrum distribution pSQI, defined in Equation (Equation 4), this equation shows how the QRS complex holds most of the information from each beat, which is concentrated in a frequency band centered at 10 Hz [17]:
(4)$$pSQI=\frac{\int_{f=5Hz}^{f=15hz}P\left(f\right)df}{\int_{f=5Hz}^{f=40hz}P\left(f\right)df}$$Additionally, this subset includes several changes to the *C*/*B*: firstly, the distance of the isoelectric line to the point found 60 ms after the J-point (by definition of the ST-segment detection), secondly distance of the isoelectric line to the point found 80 ms after the J-point (by definition of ST-segment detection), and finally the distance of the isoelectric line to the end of the R−peak.Subset *C* reuses features from subset *B*.Subset *D* reuses features from subset *B* but it considers specifically the distance from the baseline to the J-point.

The features used for the subsets were intended to keep the data processing as simple and fast as possible, to reduce the computational power to reproduce this work. The performance of the models in classification was assessed using accuracy, precision, recall, and F1 score. The resulting dataset has an imbalance of 0.82, as determined through the Shannon entropy, which can be used to measure balance, as shown in Equation (Equation 5), where *m* is the number of classes of size $c_{i}$, and *n* represents the size of the samples in the dataset. If the result is close to 0, it means an imbalance considered, whereas if the value is close to 1, the dataset is balanced. As the value in our dataset tends to 1, no further action is required to balance the classes.
(5)$$\mathrm{Balance}=\frac{-\sum_{i=1}^{m}\frac{c_{i}}{n}\log\frac{c_{i}}{n}}{\log k}$$

### 2.4. Model Development

The purpose of AutoML techniques is to obtain the optimal pipeline (defined as a sequence of data processing steps) to model the tabular dataset generated from the raw ECG data. We employed the technique proposed by us in [18]. This method seeks preprocessing steps, hyperparameter configuration, and the ML model to find the best configuration in a search space. The definition of this problem in AutoML is known as the *combined algorithm selection and hyperparameter optimization* (CASH) problem [19] and it can be defined as follows:$P=\left\{P^{\left(1\right)},\cdots ,P^{\left(n\right)}\right\}$ is a set of pipelines.$\Lambda^{\left(i\right)}=\lambda_{1}\times \lambda_{2}\times \cdots \times \lambda_{m}$ be the hyperparameter space.$\Lambda_{P}=\Lambda^{\left(1\right)}\times \Lambda^{\left(2\right)}\times \cdots \times \Lambda^{\left(n\right)}$ its combined configuration space$\lambda \in \Lambda_{P}$ a specific configuration$L(\lambda ,D_{train},D_{valid})$ the validation loss of the model created by λ, trained on data $D_{train}$ and validated on data $D_{valid}$. A set of hyperparameters was looked for, $\lambda^{⋆}$, and a pipeline $P^{⋆}$ that minimizes L. This evaluation of the loss function can be performed with k−fold cross validation [20], where $\left\{D^{\left(i\right)}\right\}$ is a subset of the dataset:
(6)$$\begin{array}{c} P_{\lambda^{⋆}}^{⋆}\in \underset{\forall P^{⋆}\in P;\;\forall \lambda^{⋆}\in \lambda }{\mathrm{argmin}}\frac{1}{k}\sum_{i=1}^{k}L\left(P_{\lambda },D_{train}^{\left(i\right)},D_{valid}^{\left(i\right)}\right) \end{array}$$

AutoML is used to solve the classification and regression tasks. We aim to predict whether an ECG chunk exhibits elevation, depression or is normal, neglecting exact numbers. We can also simplify by classifying them into normal and abnormal. In regression, we propose as output a value representing the deviation in millivolts. In the end, the two models are used as the output combined for the doctors. For regression, we use GAMA [21]. GAMA is a modular, extensible AutoML framework. In addition, models outside the GAMA search space are chosen to extend their comparison against some deep models, this choice being based on ref. [22], where they performed the comparison of regression and forecasting models.

For the classification task, we go into a little more detail. We use the generalized island model with successive halving (GEISHA), which merges successive halving [23] with bio-inspired optimization, using a generalized island architecture [24] into a different AutoML method (which uses asynchronous parallel evolution). The steps are detailed in Algorithm 1.

Algorithm 1 uses the dataset created and divides it into small parts that increase in each iteration, determined by *r*; the maximum resource *R* is the length of the dataset; and the reduction factor η defined how fast the initial number of pipelines *n* decrease exponentially over the iterations. First, $S_{max}$ is calculated, which determines the total number of iterations that occurred, better known as rungs. *T* allows *n* pipelines to be obtained from a given search space. These pipelines are evaluated with an accuracy loss function.
**Algorithm 1** GEISHA.1:**input** number of configurations *n*, sample size *r*, maximum resource *R*, reduction factor η, minimum early-stopping rate *s*2:$s_{max}=\left\lfloor \log_{\eta }\left(R/r\right)\right\rfloor$3:**assert**$n\geq \eta^{s_{max}-s}$ so that at least one configuration will be allocated R4:*T= get_hyperparameter_configurations(n)*5:**Initialize**Po←T6:**for**$i\in \left\{0,\cdots ,s_{max}-s\right\}$**do**7: Po←T This step is necessary after the lowest rung8: $n_{i}=\left\lfloor n\eta^{-i}\right\rfloor$9: $r_{i}=\left\lfloor r\eta^{i+s}\right\rfloor$10: **while not** stop_criteria **do**11:  **for**
$j\in \left\{0,\cdots ,n\_islands\right\}$
**do**12:   $Po^{\prime }\leftarrow O_{j}\left(Po_{i},\mu_{i}\right)\;$ Evaluate the population $Po_{i}$ of size $n_{i}$, with the optimizer $O_{j}$ for the task with cardinality $r_{i}$13:  **end for**14:  $M\leftarrow S_{j}\left(P^{\prime }\right)$15:  Send M to islands adjacent to $I_{j}$ in topology T16:  Let $M^{\prime }$ be the set of pipelines received from adjacent islands17:  $Po\leftarrow R(P^{\prime },M^{\prime })$18: **end while**19: $T={\mathrm{top}}_{k}\left(Po,L,n_{i}/\eta \right)$20:**end for**21:**return** best configuration in T

Following, those pipelines are considered as the population Po to be used by the optimization algorithms working in parallel. $n_{i}$ considers a small sample dataset, which speeds up the process of finding more suitable pipelines for this dataset, and $r_{i}$ represents how many pipelines are considered for the current run, since the purpose of combining SHA is to search for the most optimal solutions from those found in previous iterations. Once defined new dataset size a process of evolution was initiated, where the new population $Po_{i}$ of size $n_{i}$ was evaluated with 8 bio-inspired algorithms $O_{i}$.

M, M⊆P, and R is the migration-replacement policy, which defines the percentage of individuals that were replaced on a specific island. S defines which pipelines of the population are the best (given their loss function), which are sent to other islands; to maintain a balance between the population size, the migration-replacement policy R is used. In each rung, *T* is updated based on the best pipelines of the whole island with T topology and they are used in the next iteration. This process is repeated until all the rungs are executed.

### 2.5. Experiment Setup

Regression:-We use the default setting of GAMA, which has 50 pipelines to be optimized.-The type of optimization is done through AEA.-For the AutoML approach, the data are divided into 75% for training and 25% for testing and validation.-The metrics that are considered are mean squared error (MSE), mean absolute error (MAE), mean bias error (MBE), root mean squared error (RME) and Nash–Sutcliffe efficiency (NSE).-The time budget to find the solution is 2 h.-The models that are compared outside of the GAMA search space are: extreme learning machine (ELM), long short-term memory (LSTM), CNN, multilayer perceptron (MLP), transformers CNN, and transformers LSTM. For these models, we use the package described in [22], where in order to assess their performance, the dataset is split into 70% for training, and 30% for validation and testing, using a holdout technique as indicated in the documentation of this package.

Classification: We apply the default configuration, which consists of the following elements: replacement policy and selection policies, with 20% for each, reduction factor η=2, minimum early-stopping rate s=1, pipelines =50, and metric of negative log loss. Eight bio-inspired algorithms working asynchronously are used:-Grey wolf optimizer;-Particle swarm optimization;-Particle swarm optimization generational;-(N+1)-ES simple evolutionary algorithm;-Artificial bee colony (ABC);-Differential evolution;-Self-adaptive DE (jDE and iDE);-Self-adaptive DE (de1220 aka pDE).The data are divided into 75% for training and 25% for testing and validation.The metric considered is negative log loss.The time budget to find the solution is 2 h.

This method for AutoML can be found in the repository https://github.com/israelCamperoJurado/GAMA_generalized_island_model_AutoML.git, accessed on 24 June 2022, since then our work can be reproduced following the instruction of preprocessing and applying this optimized search algorithm [18].

Based on the best solution found in classification, we use the same features for the regression task. The results are presented as confusion matrices and a ROC curve plot for classification and scatter plot with a regression line, specifically the following:Classification in 3 classes with 4 subsets.Classification in 2 classes with specific subset and regression with same subset.

Regarding the hardware and resources requirements, the deep regression models are trained with an NVIDIA A100 Tensor Core GPU 40 GB. The AutoML approach is executed in a normal CPU with Intel Core I7-9750H, 2.60 GHz.

## 3. Results

### 3.1. Classification

First, we present confusion matrices resulting from each of the four subsets of features. Figure 4 shows a confusion matrix corresponding to classification results in the case of three classes (depressed, elevated, and normal). These results correspond to subset *A* of Table 1; we include the confusion matrix on the validation dataset, which consists of a total of 72,282 beats over the three classes. The accuracy of the three classes is 91.11%. To obtain the correct insight into the imbalancing of the dataset, we consider also the F1-score, which for this set of features is 0.8547. The class of elevated ECG beats performs poorly with an accuracy of 64.01%. However, it can also be seen that there is an overlap between the depressed and elevated class in the confusion matrix. The pipeline corresponding to this first subset of characteristics is a K-neighbors classifier, which has three preprocessing steps: impute missing values with the mean, PCA with a random singular value decomposition, and finally a fast algorithm for independent component analysis.

Figure 5 corresponds to the model trained and tested with subset *B*, where we used the exact definition of the ST-segment deviation calculation (60 and 80 mV after the J-point) together with statistical measures. We observe that the false positives and negatives when comparing the labels of the the elevated and depressed classes have decreased from 0.77% of data misclassified between depressed and elevated to only 0.61%. In addition, normal beats wrongly classified as elevated are reduced from 0.56% (subset A) to 0.08%. Moreover, the overall accuracy of the model is 96.59% in the validation dataset, as well as the F1-score of 0.95007, which indicates a better generalization in the dataset imbalance. The pipeline generated here is an extra trees classifier with two preprocessing steps: an imputer, and maximum absolute value scaler.

Figure 6 shows the results with the model trained and tested using subset *C*, in which we consider simple statistical measures combined with 3 SQIs. The overall accuracy with this model is 94.94%. It can be observed that there is a considerable improvement compared to the case in which subset *A* is used. The percentage of depressed beats mistaken for normal beats is reduced from 3.22% to 1.88%. Likewise, the elevated class with normal is improved from 0.35% of misclassified data to 0.21%. In addition, the accuracy to classify the elevated beats in this model is 83.18%, which is an improvement compared to the 64% obtained when the model is trained on subset *A*. Nevertheless, compared to the model trained and tested in subset *B*, this model displays a lower performance; considering the F1-score, this subset reached a 0.9295. This pipeline is similar to the last one, as it is an extra trees classifier with two preprocessing steps: imputer and a PCA with a random singular value decomposition.

Finally, for removing the distance at the J-80 and J-60 point and keeping only the distance from the isoelectric line to the J-point, a better result is achieved, where the last confusion matrix (Figure 7) corresponds to subset *D*, where we are only considering three simple statistical features, three sSQI, and the distance from the baseline to the J-point on the *Y*-axis, which gives us the best result with 98.30% accuracy. This subset also achieves the best F1-score of 0.9571. Although in this proposal there is an increase in misclassified data among the elevated class, it can also be seen that the normal class has almost perfect performance; likewise, the depressed beats considered as normal are reduced to 0.09%. This last pipeline is again an extra trees classifier with two preprocessing steps: an imputer, and a filter to select the *p*-values corresponding to the family-wise error rate.

As can be seen in the confusion matrices, the number of samples is changing through the approaches; this is due to the method of precomputing the features for each dataset. In practice, given the functions for preprocessing, it can happen that not all the features for a specific beat are identified; if that is the case, we discard such a peak. The final dataset is described in a better way in Table 2, which includes the statistical information concerning the features and the label of the dataset.

One of the main points in this work is the interpretability of our approach compared with CNN and deep models. Since the extra trees classifier is the model with the best performance through the preprocessing and optimized search, it is possible to retrieve the estimators (decision trees) and interpretable rules nested on it to create a white box model based on simple decisions. Considering an ECG time series input, it is only necessary to preprocess each beat, as we explain beforehand to build the dataset and apply the rules that can be found in the next URL together with the full tree diagram in PDF format; these rules are not integrated into this paper, for space and organizing convenience https://github.com/israelCamperoJurado/GAMA_generalized_island_model_AutoML/blob/master/ST_segment_paper_rules/, accessed on 24 June 2022.

In addition, the multiclass problem divided into normal, elevated and depressed is handled as a binary problem; the purpose is to see the improvement of generalization by considering only two classes since many of the works in the state of the art focus on the identification of normal and abnormal beats. As the feature set is already established, elevated and depressed beats are considered as one class versus normal. See Figure 8, the ROC plot of our implementation, which represents a 99.87% accuracy considering only normal beats and with some deviation.

### 3.2. Regression

For the regression part, the results of the models that are not in the GAMA search space are included first. See Table 3. where the results in MSE and mean absolute error (MAE) for each of the methods. As shown in the table, the model with the best generalization over the points of the dependent variable is ELM, which presents an MSE of 0.030534.

Given that we have selected the subset of characteristics that were best adopted, we proceed to use them for regression. For the included models, we have a scatter plot, as well as plotting the LR to represent the ratio of the linear regression against the actual values; 1:1 is the line where measure=predicted, together with graphs representing the residuals in each model. The first results from models that are not in GAMA search space are included. As shown in Figure 9 (scatter plot Figure 9a, model residuals Figure 9b) MLP performs relatively well with RMSE = 0.2045 mV. However, the 1:1 line which tries to fit the actual values to the measured values is not ideally coupled to LR.

For the ELM, Figure 10 (scatter plot Figure 10a, model residuals Figure 10b), it is observed that RMSE = 0.1747 mV. In terms of MBE, the model performs similarly to the MLP. On the other hand, the NSE has a strong variation, going from a negative to a positive value. This last result suggests that the ELM is a better predictor than the average.

In the case of LSTM, in Figure 11 (scatter plot Figure 11a, model residuals Figure 11b) the MBE = 0.1192 mV is slightly worse than ELM. Additionally, the regression line fits less accurately the measured data compared to the obtained data. The 1:1 line also shows a loss in the generalization of information.

On the other hand, in the CNN, Figure 12 (scatter plot Figure 12a, model residuals Figure 12b), we have a better fit of the 1:1 line; likewise an MBE = 0.078 mV is presented. Unfortunately, the NSE value is decreased again, indicating that the model has problems predicting ST-segment variation.

Now presenting the graphs of the transformer-based models, for the combination with LSTM, Figure 13 (scatter plot Figure 13a, model residuals Figure 13b), similar values are obtained to those obtained with ELM, the problem again is the NSE, which is decreased below zero more than the normal LSTM.

The other transformer, Figure 14 (scatter plot Figure 14a, model residuals Figure 14b), which is based on CNN, also presents a negative NSE, as well as a higher MBE than the transformer with LSTM.

Table 3 contains the MSE and MAE for each of the methods. As shown in the table, the model with the best performance over the points of the dependent variable is ELM, which presents an MSE = 0.0305 mV. However, speaking of MAE, the best is transformers CNN with 0.1290 mV, almost identical to CNN 0.129487 mV.

Considering the results for these models against the GAMA optimized search space, see Figure 15 (scatter plot Figure 15a, model residuals Figure 15b), it can be seen that the regression line fits better than the previous models on the actual values. The MSE we obtain is 0.0076, which is approximately four times lower than ELM, which was the best of the previous models. Likewise, other advantages are that the NSE is not only positive, but it is in a range of 0.7143, quite close to 1. Another good indicator is $R^{2}$, where the value obtained is approximately equal to 0.8. Similar to the classification task, the winning model corresponds to a regressor extra trees with a simple imputer and a PCA with a random singular value decomposition.

Examples of our results can be seen in Figure 16, detailing the statistical and signal characteristics, as well as identifying the most relevant points of the signal and providing the corresponding classification and regression output.

Looking for a point of comparison, the transformer LSTM took approximately 90 min to train, and the AutoML approach took exactly two hours, which is the limit time that was defined. Since the NVIDIA A100 ups to 245 times higher AI inference performance over CPUs (relative performance in sequences per second) [25] we computed how much lighter our approach is in terms of computational resources needed. This can be seen in Equation (Equation 7).
(7)$$\begin{array}{c} \mathrm{NVIDIA}\;A100\to 245\;\mathrm{CPUs}\\ 245\times (90\times 60)=1323000\;(\mathrm{sequence}\;\mathrm{per}\;\mathrm{second}\;\mathrm{required}\;\mathrm{for}\;\mathrm{GPU})\\ 120\times 60=7200\;(\mathrm{sequence}\;\mathrm{per}\;\mathrm{second}\;\mathrm{required}\;\mathrm{for}\;\mathrm{CPU})\\ 1323000/7200\to 183.75\;\mathrm{less}\;\mathrm{power}\;\mathrm{required} \end{array}$$

## 4. Discussion

### 4.1. Main Findings

There is an improvement in the classification of elevated versus depressed segments when statistical features are considered. This indicates that the simple statistical characteristics of a signal may intrinsically represent its morphology, as is the case. This also allows us to suggest that in the presence of ST-segment deviation in one beat of the ECG, the morphology of the whole beat changes and presents more pronounced peaks in the other waves. It can be seen that the false positives and negatives between the elevated and depressed classes are decreased due to the inclusion of guiding features to generally identify the structure of the beat.

When features such as 60 and 80 mV after the J-point are used in conjunction with the statistical measure, the false positives and negatives between the elevated and depressed classes are decreased. This combination of features shows that information about a signal and its morphology can be extracted by simple statistical measures, and since the J-point is the inflection point that serves as a reference to the ST-segment, it provides indispensable information for correct identification.

It is possible to notice not only the influence of the SQI or statistical variables, but also the selected model, and more specifically the preprocessing steps. When the accuracy is 91%, a K-nearest neighbors is used, as it is known that this keeps a perspective of each sample point in the dataset to associate new entries with the closest one. However, the performance of the model starts to improve drastically when using extra trees classifier, where the difference is the processing methods; we can see that when using the maximum absolute value scaler, the accuracy is considerably higher than when using a PCA, so we can infer that it is better to keep the initial variables than to create new ones. Finally the best preprocessing step is filter to select the p-values corresponding to the family-wise error rate, which means that an internal test is performed to identify significant differences, and this allows to clearly discriminate between the ECG beat classes.

When AutoML is used in tasks properly, it can outperform deep model results. An extra trees classifier optimized through bio-inspired algorithms and multi-fidelity processes outperforms previous work. Since, to the best of our knowledge, there are no works that focus on the regression task to predict the millivolt values of the ST-segment deviation, we perform our own comparison, where an extra trees regressor produces an NSE value approximately 14 times better than the ELM and even better than the transformer CNN and transformer LSTM. This is a good indicator, as it indicates whether a model is a better predictor over new values than the mean itself. Additionally, the $R^{2}$ allows us to see that the variance over our dependent variable is well contained over the seven features we use to model.

### 4.2. Related Research

There are several comparable publications [10] that worked with exactly the same database as us; however, our aim is not to characterize different CVD, but if we consider only the ST-segment deviation modeling results, they obtained 97.89% 90.92% and 94.28% in precision, recall and F1, respectively. Considering the binarization of the problem, we obtained 98% for the three metrics in weighted average, and 97%, 94% and 96%, respectively, for the macro average when considering the multiclass problem. So our approximation is more accurate than theirs and lighter since we do not work with deep models.

Regarding [12], they had an excellent result in MI detection over 12-lead ECG signal diagnosis, even though we are not using the same database or the same problem. MI is highly related to ST-elevation; therefore, in future work, we plan to perform the modeling of such a disorder and compare more objectively this work. In [13], the authors used the same database and also had an excellent result with an accuracy of 99.23%; being more exact, they obtained 97.8% sensitivity, 100% specificity for the detection of ST-segment changes, against 99.11% of us 94.3% respectively. However, we are considering three classes and not only the division between normal and abnormal. Therefore, given that our work has a better performance in sensitivity, our approach is more powerful for detecting true positives but can still improve for detecting true negatives, as in specificity, we do not outperform the cited work.

The reason why our model has a good performance has two points. The first one is due to the search space, as it is known that a good search space can lead to good solutions, even if the search optimization method is random. Similarly, the optimization method used, SHA, allows, even when the dataset contains more than 70 K samples, to handle that information quite easily. Combining these two points generates a much lighter model than neural implementations, which in turn allows us to interpret the output thanks to the deviations we obtain from the isoelectric line to the baseline and the SQIs.

As can be seen previously, one of the variables that has a significant difference for the characterization is the distance from the isoelectric line to the point of inflection J, as well as sSQI, where the latter is well known for representing the asymmetry or distribution of our signal, which allows us to obtain a notion of stability to identify a normal beat and one that is not. Future work is the validation of this algorithm with real patients in day-to-day activities. For this, it will be necessary to subtract ECG segments in moments of physical or psychological stress, which are the moments when such abnormalities are usually found.

## 5. Conclusions

To conclude, we can say that we provided a tool for ST-segment deviation detection with an accuracy of 99.87% when the problem is binarized and 98.30% when considering the three classes defined at the beginning. This tool allows us to view the quality of the ECG beats independently. Our implementation differs from the rest in the sense that our model is much lighter, is optimized through state-of-the-art techniques and is interpretable due to the interpretations we give on the distance metrics on each ECG beat, together with the statistical characteristics and SQIs.

In future work, we will implement this work with online learning to be able to handle stream data. Since the goal is to use this algorithm in real time for patients and athletes monitoring, we must determine drift detection and retrain the algorithm when required.

## Figures and Tables

**Figure 1 sensors-22-04919-f001:**
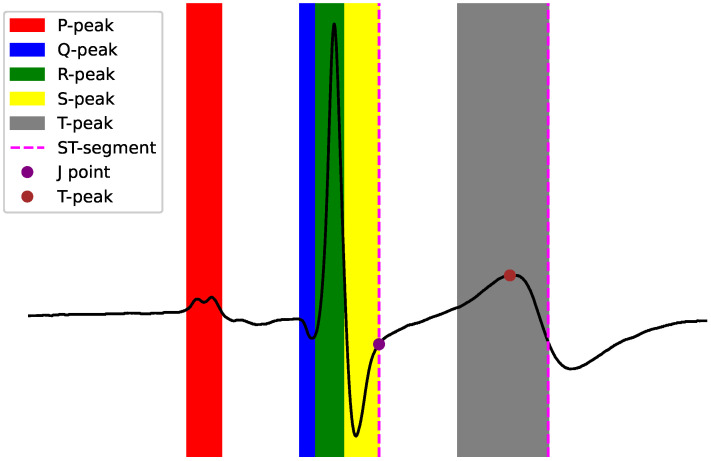
ECG beat morphology example. Note the vertical dotted lines representing the ST-segment, which is the interval between ventricular depolarization (shift in electric charge distribution inside the cell) and repolarization (J-point elevation) [4].

**Figure 2 sensors-22-04919-f002:**
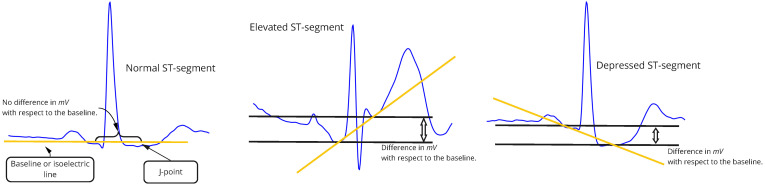
ST-segment deviation visual, the traditional way to identify this abnormality by measuring the distance between the ST-segment and the isoelectric line. Normal ECG beat (**left**), elevated ST-segment (**center**), depressed ST-segment (**right**).

**Figure 3 sensors-22-04919-f003:**
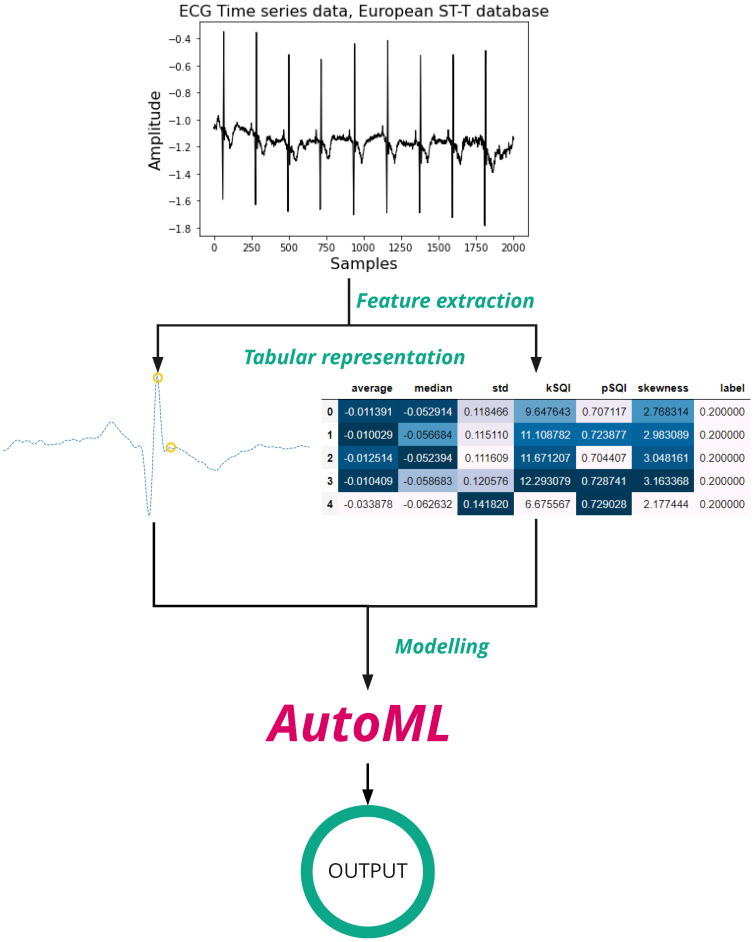
Overview of the process to the identification of ST-segment deviations.

**Figure 4 sensors-22-04919-f004:**
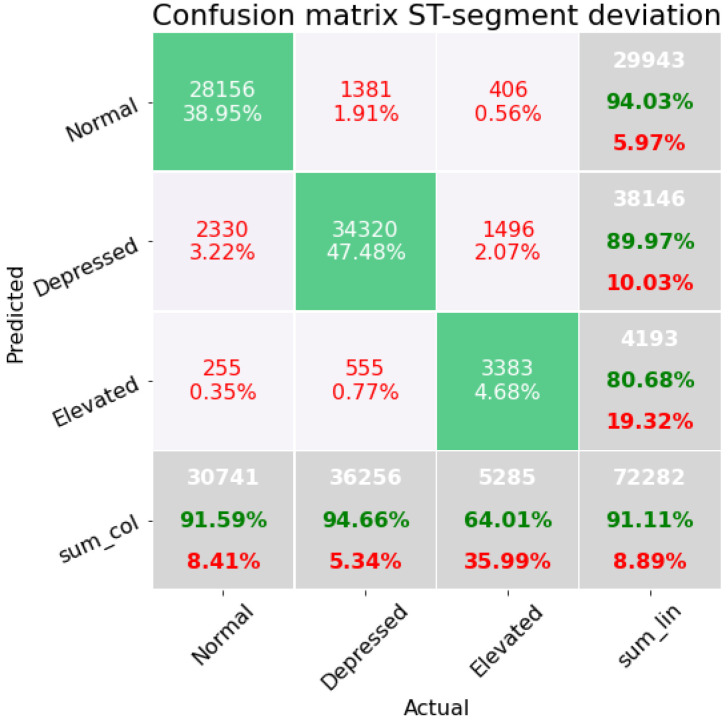
Confusion matrix on the modeling containing the features concerning the distances between the waves of the ECG beat.

**Figure 5 sensors-22-04919-f005:**
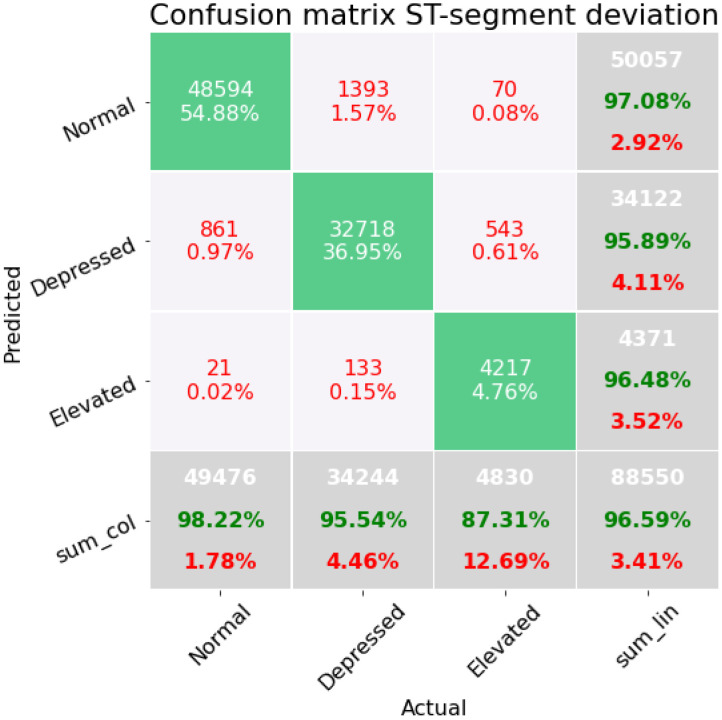
Confusion matrix on the modeling containing the features concerning statistical measurements, SQIs of the ECG beat and distance from the baseline at two different points after the J-point, 60 and 80 mV respectively.

**Figure 6 sensors-22-04919-f006:**
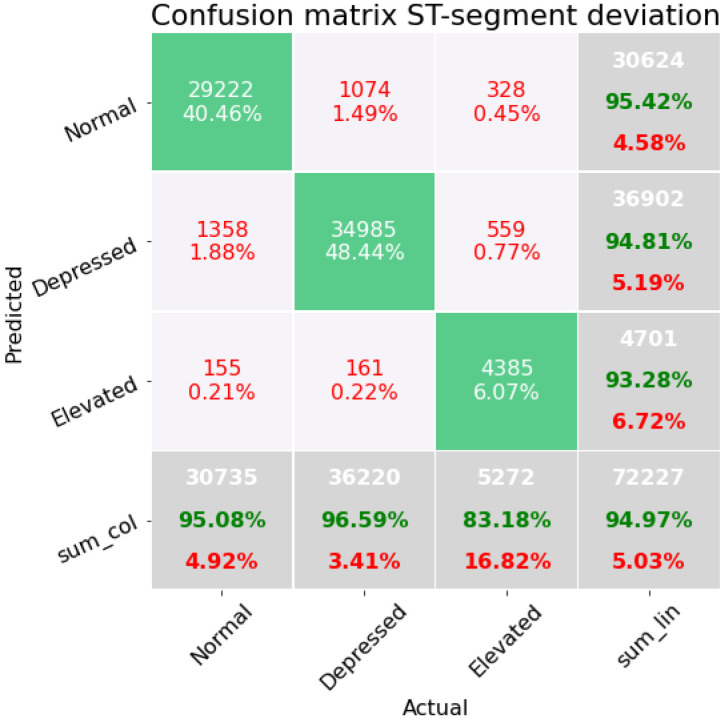
Confusion matrix on the modeling containing the features concerning statistical measures, distance from the baseline to J-point, and SQI of the beat ECG.

**Figure 7 sensors-22-04919-f007:**
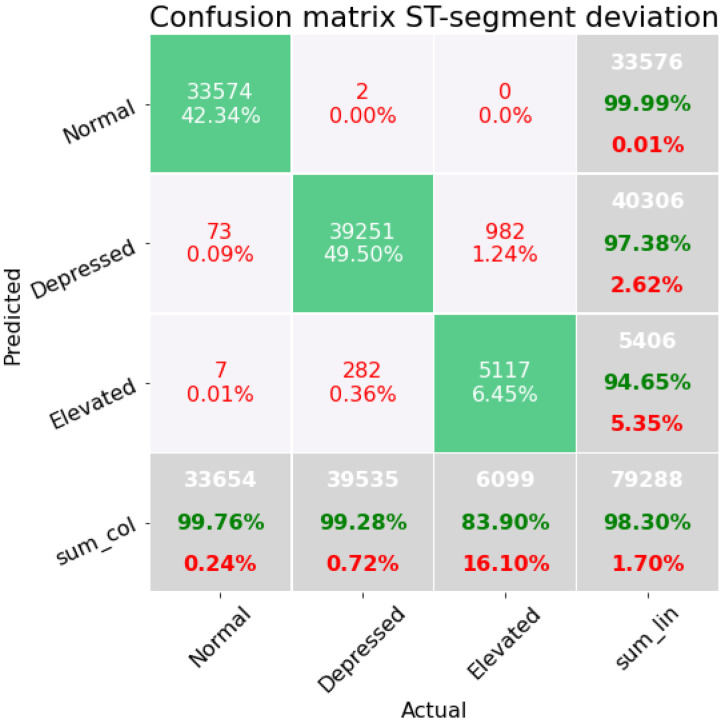
Confusion matrix on the modeling containing the features concerning statistical measures and SQIs of the ECG beat.

**Figure 8 sensors-22-04919-f008:**
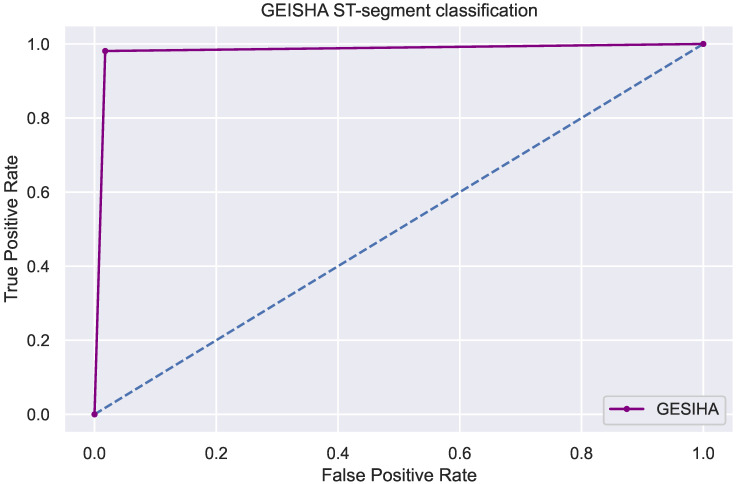
The area under the curve with binarization of the problem, considering only normal and abnormal beats (depression or elevation as one).

**Figure 9 sensors-22-04919-f009:**
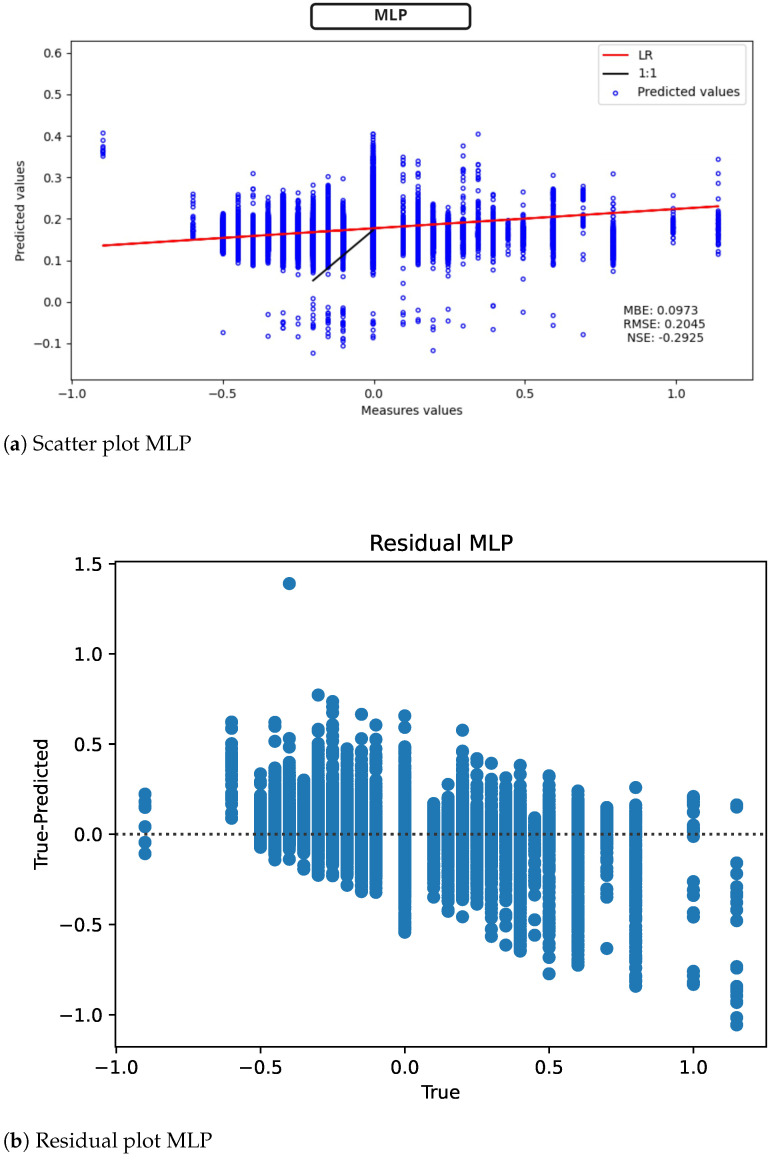
MLP results for ST deviation calculation in ECG time series.

**Figure 10 sensors-22-04919-f010:**
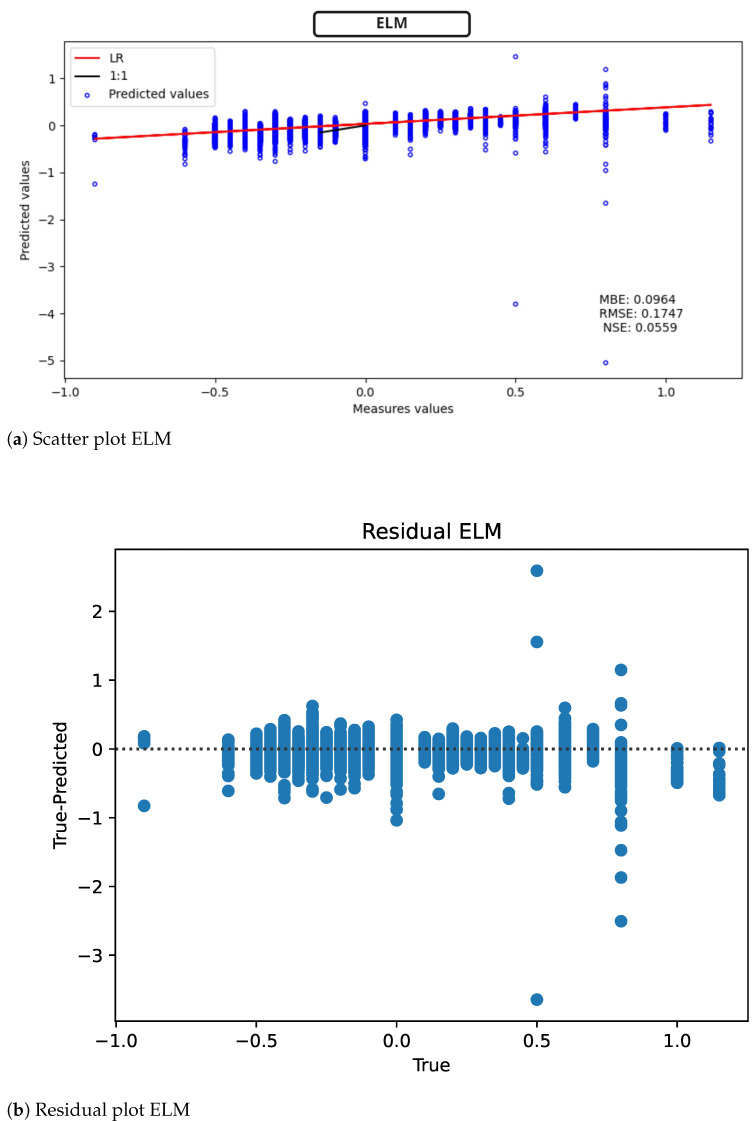
ELM results for ST deviation calculation in ECG time series.

**Figure 11 sensors-22-04919-f011:**
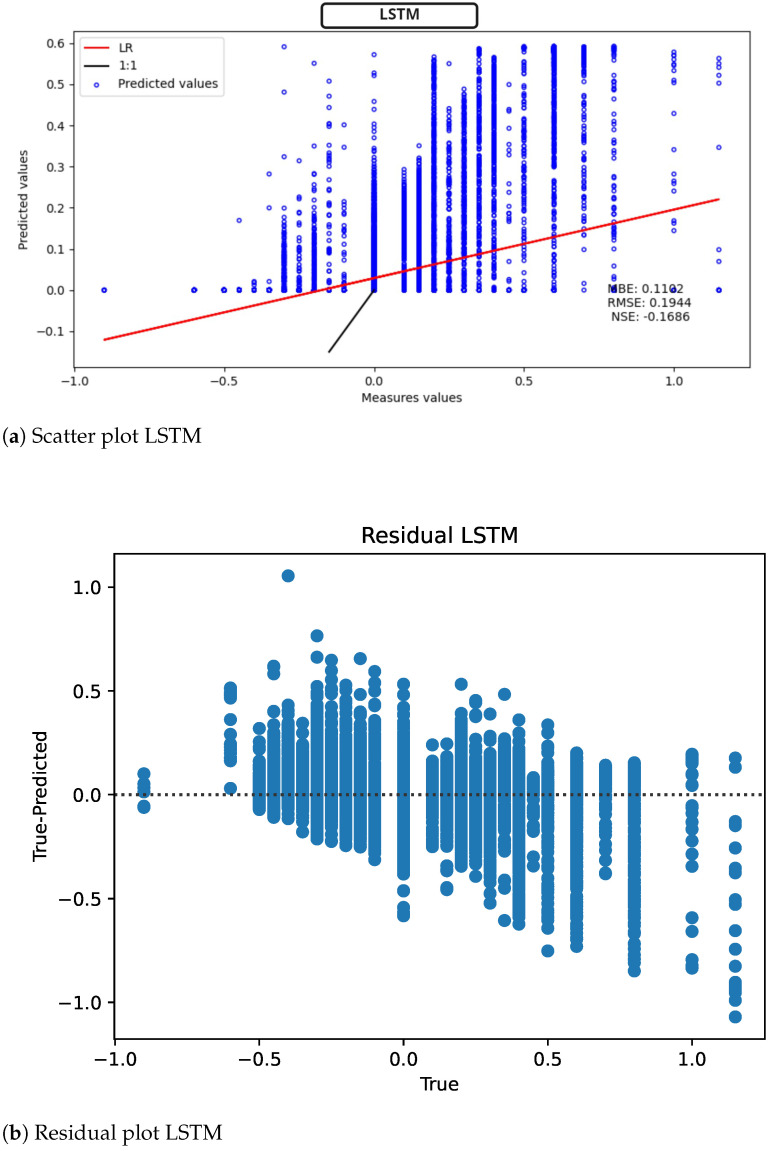
LSTM results for ST deviation calculation in ECG time series.

**Figure 12 sensors-22-04919-f012:**
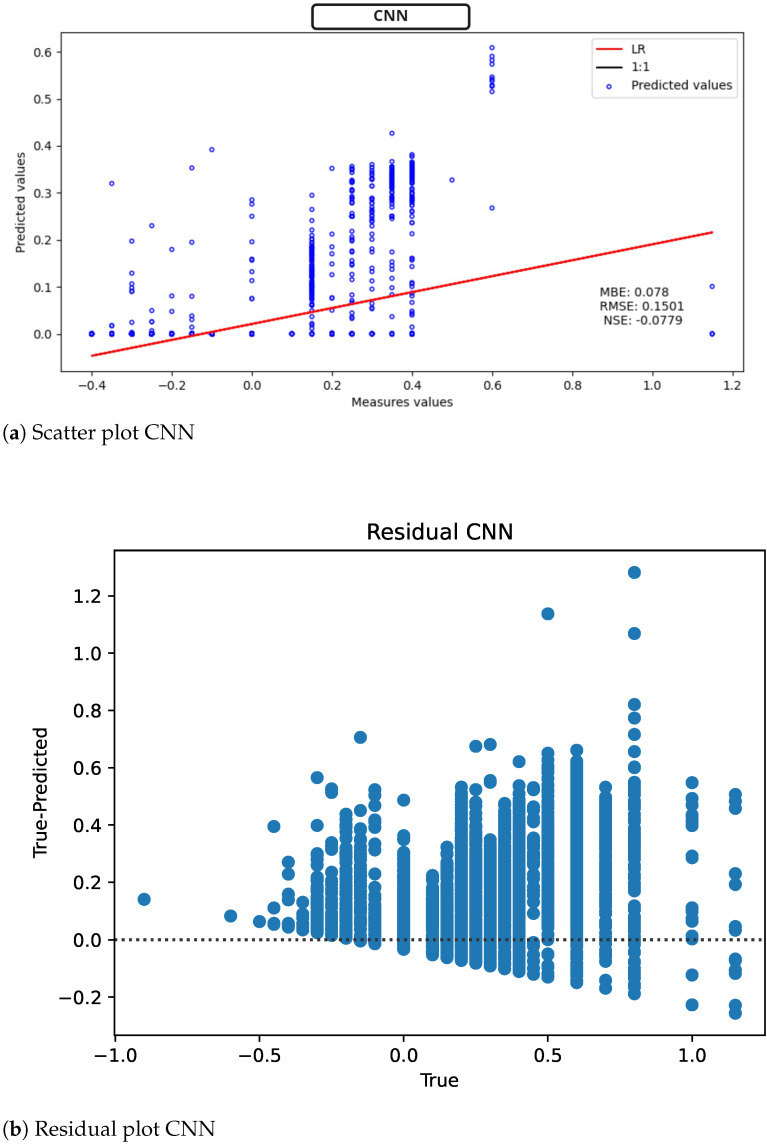
CNN results for ST deviation calculation in ECG time series.

**Figure 13 sensors-22-04919-f013:**
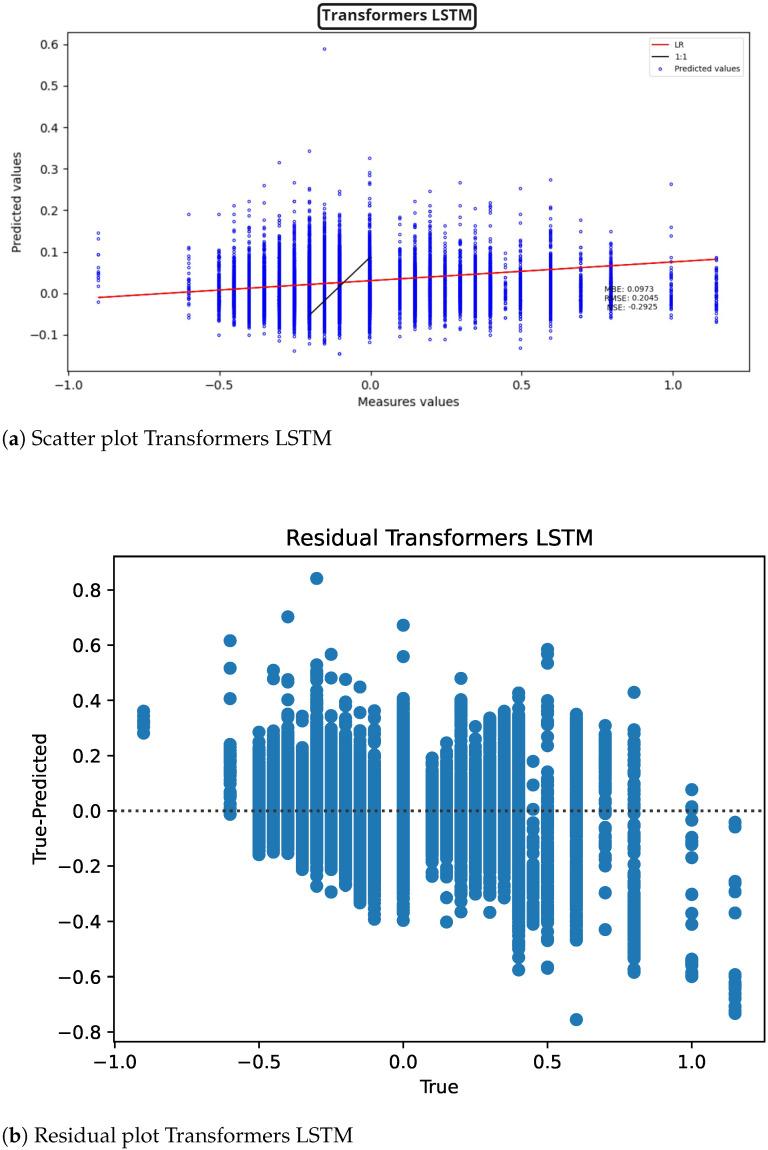
Transformers LSTM results for ST deviation calculation in ECG time series.

**Figure 14 sensors-22-04919-f014:**
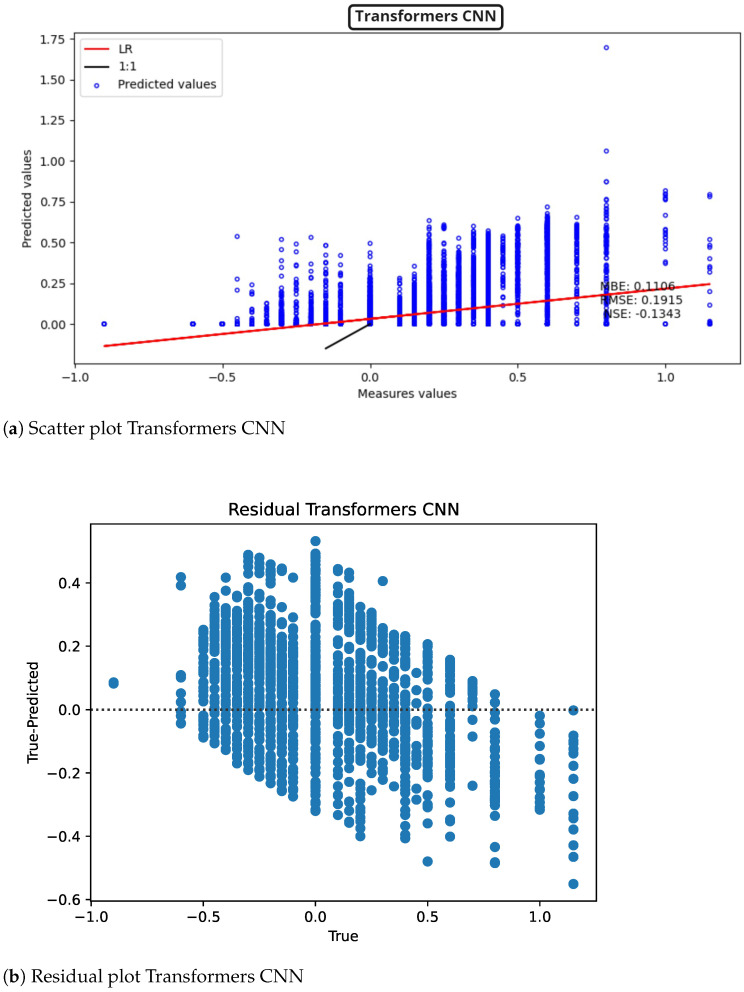
Transformers CNN results for ST deviation calculation in ECG time series.

**Figure 15 sensors-22-04919-f015:**
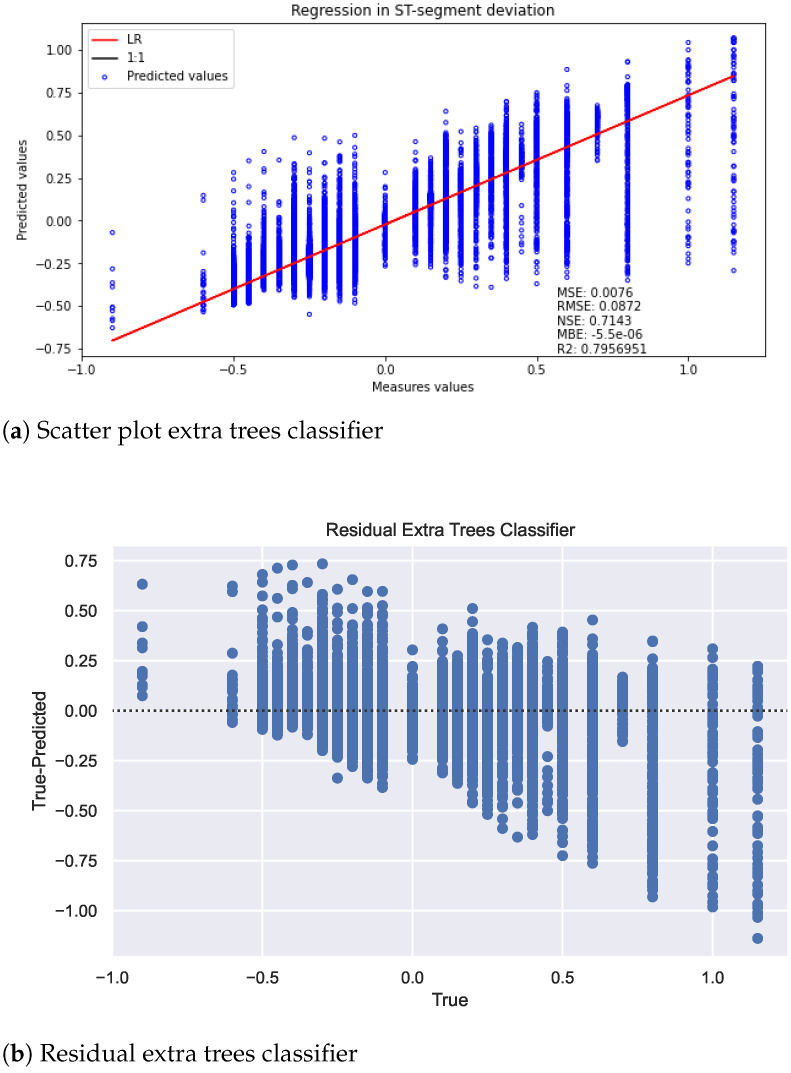
Result in the regression task used for this approach.

**Figure 16 sensors-22-04919-f016:**
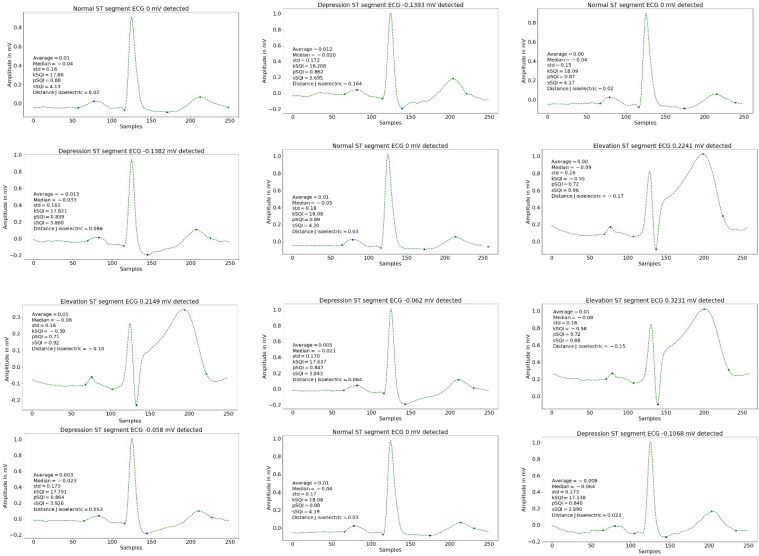
Result in the regression task used for this approach. Detection of several ECG beats, some of them categorized as normal, depressed or elevated.

**Table 1 sensors-22-04919-t001:** Different subsets of features used for prediction. *A*: corresponds to the use of distance metrics between ECG beat peaks, *B*: corresponds to the use of statistical measures SQIs and distance metrics, *C*: only considers statistical metrics and SQIs, *D*: has a combination of statistical measures, SQIs and distance measures at two points after the J-point.

Subset	Features
*A*	Time–distance (millisecond) from the R-peak to: to P-peak, Q-peak, S-peak and T-peak. Time–distance from R peak to P onset, P offset and T offset.
*B*	Average, median, std, kSQI, pSQI, sSQI, distance from the baseline to 80 ms after the J-point, and distance from baseline to R offset.
*C*	Average, median, std, kSQI, pSQI and sSQI.
*D*	Average, median, std, kSQI, pSQI, sSQI and distance from the baseline to J-point.

**Table 2 sensors-22-04919-t002:** Description of the final dataset generated by selecting statistical characteristics and the distance to point-J.

	Average	Median	STD	kSQI	pSQI	sSQI	Distance J-Isoelectric	Label
**mean**	−0.00053	−0.021430	0.1936990	8.710861	0.7693565	1.165187	0.13038	0.6474574
**std**	0.02352	0.051486	0.0913441	6.670272	0.1121232	2.197134	0.207263	0.6117591
**min**	−1.05441	−2.004105	0.0071646	−1.875964	0.0927229	−7.886973	−2.58223	0.0
**25%**	−0.00818	−0.0453055	0.1341400	2.930704	0.7013930	−0.454002	0.023201	0.0
**50%**	0.000163	−0.0163238	0.1810513	7.748890	0.7742705	1.4952312	0.1166997	1.0
**75%**	0.007835	0.0085502	0.2439483	13.43589	0.8516178	2.9652297	0.2589208	1.0
**max**	1.416646	3.1385591	4.3804516	65.90857	0.9986373	6.4377815	3.568406	2.0

**Table 3 sensors-22-04919-t003:** Regression models of a different search space for comparison against the proposed analysis.

Model	MSE	MAE
*ELM*	0.0305	0.1335
*LSTM*	0.0377	0.1316
*CNN*	0.0368	0.1294
*MLP*	0.0418	0.1387
*Transformers CNN*	0.03668	0.1290
*Transformers LSTM*	0.0418	0.1387

## Data Availability

The database we used is public but the entire project cannot be shared due to a confidentiality agreement in the project.

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
