# Peer review of "Interpretable Assessment of ST-Segment Deviation in ECG Time Series"

_sensors, 2022, doi:10.3390/s22134919_

Round 1

Reviewer 1 Report

The manuscript by Israel Campero Jurado entitled "Interpretable assessment of ST-segment deviation in ECG time-series" presents a method to classify ST-segment deviation. The feature extraction and data modeling are done automatically by means of an automated machine-learning algorithm. The proposed algorithm allows the identify normal, elevated, or depressed ECG beats. The authors also propose a regression algorithm to estimate the deviation value in mv. The proposed algorithms are tested using an international database and the results show high performance. The current work has been documented with rigor, it is well written, and the results seem to be significant progress over the state of the art, but some points have to be clarified before a final suggestion.

-Please increase the size of the figures, particularly, figure 3.

- I understand that the data cannot be shared due to a confidentiality agreement in the project. However, it would be good practice to provide some sample of its automated machine learning method on GitHub. Otherwise, I feel, there is a big hurdle that other groups will pick up the approach, presented. But that is the sole purpose of publishing this work.  So I would encourage the authors to provide some sample code along with the paper.

- A quantitative description of the dataset is an important topic that was omitted. E.g., what is the number of samples (batch size) used for the training, test, and validation phases?

-In the confusion matrix of Fig 4, the number of samples for class is unbalanced, i.e., there are 30741 samples for “normal” class, 36256 for “depressed” class, and 5285 for “elevated” class. It is known well an unbalanced dataset can bias the model. This impacts dramatically the ability of the algorithm to predict, i.e., the algorithm could predict “benign” for all samples and still gain a very high accuracy. In this case, one obtains an apparent "good" accuracy. What do you make to manage the challenges of imbalanced class data?

Reviewer 2 Report

The contribution of this study is a method to classify ST segment deviation and detect its abnormality in mV.

The output of the algorithm is a classification of ECG beats into normal, elevation, or depressing classes.

Given a time series of ECGs, it isolates them beat by beat and extracts detailed characteristics to form a tabular data set.

This new representation was used to perform the tasks of regression and classification by the ML model through AutoML to optimize the results.

The use of features such as 60 mV or 80 mV after the J point in combination with statistical measures shows that information about the signal and its morphology can be extracted by a simple statistical measure.

The achievement of a regression task to predict the mV value of ST segment deviations to be judged to show its research originality and new findings.

Of the many contributions, the following two require additional explanation.

1. the definition of "interpretability," which is also included in the title of the paper, is unclear.

In the introduction, the paper mentions interpretability by citing references using CNNs.

In the Conclusion, since the interpretation is given to the distance metrics for each ECG beat along with the statistical properties and SQI.

However, if the relationship between the inputs and the outputs is not shown, then it cannot be considered interpretable, can it?

What is interpretability in your opinion?

2. you conclude that the required computational power is reduced.

While this is qualitatively understandable, we recommend that an evaluation of how much reduction is quantitatively achieved be made.

Please confirm the following as well, which may require minor modifications.

- In the legend of Figure 1, the gray area should be T-peak, not S-peak.

- In line 44, the phrase "60 / 80 milliseconds from the J-point" should be mV. Also check Subset B in Table 1. If these are correct, check to see if the mV in lines 128 and 129 are correct.

- In line 124, the formula number is not correctly referenced.

- Figure 8 and the paragraph beginning at line 268 describing this are not clear.

- In line 292, please correct MV to mV.

- Reference 18 is not appropriate because it is unpublished.
